# Breeding Strategies to Optimize Effective Population Size in Low Census Captive Populations: The Case of *Gazella cuvieri*

**DOI:** 10.3390/ani11061559

**Published:** 2021-05-27

**Authors:** Candela Ojeda-Marín, Isabel Cervantes, Eulalia Moreno, Félix Goyache, Juan Pablo Gutiérrez

**Affiliations:** 1Departamento de Producción Animal, Facultad de Veterinaria, Universidad Complutense de Madrid, Avda. Puerta de Hierro s/n, E-28040 Madrid, Spain; candelao@ucm.es (C.O.-M.); gutgar@vet.ucm.es (J.P.G.); 2Estación Experimental de Zonas Áridas-CSIC, 04120 La Cañada de San Urbano, Almería, Spain; emoreno@eeza.csic.es; 3SERIDA-Deva, Camino de Rioseco 1225, E-33394 Gijón, Spain; fgoyache@serida.org

**Keywords:** conservation, small populations, effective population size, mating designs

## Abstract

**Simple Summary:**

Small-sized populations frequently undergo a significant loss of genetic variability that can lead to their extinction. Therefore, research on animal breeding has focused on mating systems for minimizing the disappearance of genetic variability. Minimizing the average coancestry of offspring has been described as the best strategy for this purpose. Traditionally, the preservation of genetic variability has been approached via breeding strategies for increasing the effective population size (Ne). The main objective of this study was to compare, via computer simulations, the performance of different breeding schemes to limit the losses of genetic diversity in small populations. This objective was achieved by monitoring the evolution of the effective size obtained by different strategies across 20 generations with a starting point of two pedigree real populations of *Gazella cuvieri*. The results showed that minimizing average coancestry in a cohort did not maximize the effective size as compared with new strategies that were designed for this study. Furthermore, the best strategy may vary for each population and should be studied individually.

**Abstract:**

Small-sized animal populations can undergo significant loss of genetic variability that can lead to their extinction. Therefore, studies on animal breeding have focused on mating systems for minimizing the disappearance of genetic variability. The main objective of this study was to compare, using computer simulations, the performance of different breeding schemes to limit the loss of genetic diversity in small-sized populations. This objective was achieved by monitoring the evolution of the effective population size obtained by 23 strategies throughout 20 generations in two populations of *Gazella cuvieri*. The scenarios were designed with different assumptions, in both reference subpopulations, regarding: the use of parents coancestry or offspring coancestry, the use of their increases or the coefficients themselves, and the number of males and females involved. Computations were performed using an experimental module of Endog v4.9 developed for this purpose. The results of the study showed that strategies for minimizing the coancestry of the parents were better in the short term; however, these strategies were worse in the long term. Minimizing the average coancestry of the offspring was a better approach in the long term. Nevertheless, in both populations, the best results were obtained when both the coancestry of the parents and the coancestry of the offspring were weighted at 5% each and neither males nor females were assumed to contribute to the next generation. In any case, not all strategies had the same evolutionary pattern throughout generations in both populations. The current results show that neither traditional nor new strategies have any general use. Therefore, it is important to carefully test these strategies before applying them to different populations with different breeding needs under different conditions, such as different generation intervals, and different natural breeding systems such as monogamy or polygyny.

## 1. Introduction

When dealing with small-sized populations, studies on animal breeding have concentrated on mating strategies to minimize the loss of genetic variation [1]. Nevertheless, in large breeds with a small effective population size, the focus has been on maximizing genetic gain under artificial selection pressure, whilst minimizing inbreeding and loss of genetic variation [2,3,4]. However, preserving genetic diversity in a captive population is itself an issue. In scenarios of critically endangered species, protection provided by reservations or zoos may be increasingly necessary for short- and long-term survival [5,6].

Preservation of genetic variability has traditionally been achieved by breeding strategies aimed at increasing the effective population size (Ne) and, because of their inverse relationship, minimizing the rate of inbreeding (ΔF) [7]. Studies are in agreement that minimizing coancestry (f) in a cohort [8] outperforms other breeding strategies [9,10,11,12]. The expected heterozygosity (He), i.e., Nei’s gene diversity, is the complementary value of coancestry (He = 1 − f), and therefore minimizing f in a cohort maximizes He [5]. Moreover, in pedigrees with uniform coancestry between individuals, this method should converge with the traditional method of minimizing the variances of family size [13,14].

However, in short-term breeding schemes, minimizing the coancestry between parents may slightly outperform schemes minimizing the mean f in the offspring cohort. Minimizing coancestry in a cohort causes deceleration of inbreeding accumulation in the first few generations via the appearance of population structuring, i.e., the formation of inbred families [9,15]. After the inbred subpopulations are formed, the loss of diversity due to the absence of mendelian segregation leads to a dramatic decrease in population fitness and an increase in the probable extinction of the target conservation population [9,10]. Even if structuring is successfully created, the sudden extinction of one of the inbred subpopulations implies an irreparable loss for the viability of the population. It has been suggested that the performance of different breeding strategies in small-sized populations can be assessed by comparing the deviation from Hardy–Weinberg proportions [10]. In non-random mating populations, there may be a longer delay between coancestry and inbreeding, as can be seen in F_IS_ statistics [7,16]. However, in real small-sized populations it is impossible to avoid mating close relatives. This is even true in simulated scenarios because of side effects relating to the influence of the mating design at the selection phase [9].

Although there are many reports about evaluating breeding schemes designed for preserving genetic variability in small-sized populations [3,9,10,13], it is still worthwhile revisiting the subject. New methods based on individual increase in inbreeding [17,18] or in coancestry [19] have been developed to estimate effective population size and, at least in real pedigrees, they have shown to be more flexible and accurate for assessing effective population size [17,20]. Moreover, from a practical point of view, assessment of breeding schemes should raise some concerns, namely: (a) the starting point of the simulations hardly includes only unrelated individuals; (b) the genetic scenario may vary between populations at the time of starting a preservation plan due to different breeding histories, this fact should be carefully considered before implementing a conservation program; and (c) breeding schemes dramatically vary whether or not all reproductive individuals (or, at least, all available males) are assumed to contribute an offspring to the next generation.

The present study compares computer simulations of the performance of different breeding strategies aimed at limiting the loss of genetic diversity in small-sized populations. This is assessed via fluctuations of the effective population size, in both the short and the long term. The methods were applied on two different scenarios from the pedigree of the *Gazella cuvieri*, a Sahelo-Saharan species classified as vulnerable by the IUCN (International Union of Conservation of Nature) mainly due to its small population size and for which an ex situ endangered program (EEP) has been run in captivity, since 2006 [21,22].

## 2. Materials and Methods

### 2.1. Study Species

Cuvier’s gazelle is a medium-sized species whose populations have steeply declined since the 1950s, apparently due to excessive hunting and habitat degradation in their range (Morocco, Tunisia, and Algeria) [23]. The most recent estimates of its whole population suggest that, currently, it is between 2360 and 4560 individuals, most of which are in Morocco [24]. It is sexually dimorphic, adult males are 24% heavier than adult females, and the average body mass of adult females is 26 kg, and for adult males it is 34 kg [21]. Individuals live mostly in small groups of 5 to 6 [25,26]. According to its polygynous mating system, in autumn, harems are conformed, and made up of one adult male and several females (3–6) accompanied by their yearlings. During the rut, young males are forced to leave their maternal herds and gather in bachelor groups. They may, subsequently, be joined by males evicted during fights over females [23]. Once formed, the harems remain together all winter and only separate when the females leave to give birth in the spring. Gestation is about 5.5 months and twins represent up to 39% of births [21].

As a starting point, a simulation study was carried out using two subsets of the real pedigree of the *Gazella cuvieri* populations located in Almería (Andalusia, Spain) and in La Lajita (Canary Islands, Spain). The reference population of the Almería flock consisted of all the animals born between 2014 and 2015 (16 females and 6 males). The reference population of the La Lajita flock was born between 2014 and 2015 (11 females and 8 males).

The generation interval was calculated for the whole pedigree, defined as the average age of parents at the birth of their progeny kept for reproduction [27]. Furthermore, the average of individual inbreeding, defined as the probability that an individual has two identical alleles by descent [28], was calculated for both populations and within males and females in each population. In addition, equivalent complete generations (EG) were computed as the sum of overall known ancestors of the terms computed as the sum of (1/2)*^n^*, where *n* is the number of generations separating the individual to each known ancestor [29]. Finally, the effective population size, defined as the number of breeding animals that would lead to the actual increase in inbreeding if they contributed equally to the next generation, was assessed via individual increase (NeF) [18] of inbreeding and increase in pairwise coancestry (NeC) [19] for both populations.

Different mating strategies were tested by minimizing a function *Fx* in which the number of males were assumed to be equal to or lower than the number of females, thus, maintaining a similar sex ratio throughout the different generations. The strategies were classified into three different groups which focused on offspring inbreeding or mating coancestry addressed below as parents’ coancestry (Section 2.2) or offspring coancestry (Section 2.3) or a combination of both (Section 2.4).

### 2.2. Strategies to Minimize the Parents’ Coancestry


For minimum coancestry (F), $Fx=\sum^{}C_{jk}$, where *C_jk_* is the coancestry between male *j* and female *k*. In this strategy, all the females were involved in the solution. Two alternatives were checked:Strategy F0, i.e., not all males were expected to contribute offspring to the next generation;Strategy F1, i.e., all males were expected to contribute offspring to the next generation.For minimum increase in coancestry (ΔF)$,\;Fx=\sum^{}\Delta C_{jk}$, where $\Delta C_{jk}=1-\frac{g_{j}+g_{k}}{2}\sqrt{1-C_{jk}}$ [15], with *g_j_* and *g_k_*, being the equivalent discrete generations known for individuals *j* and *k* [29,30]. Since coancestry accumulates per generation, under this strategy increases in coancestry were corrected for individual differences in pedigree depth. Again, there were two alternatives:Strategy ΔF0, i.e., not all males were expected to contribute offspring to the next generation;Strategy ΔF1, i.e., all the males were expected to contribute offspring to the next generation.The strategy for minimum coancestry free (Ff) was similar to Strategy F1, but, in this case, not all the females were expected to contribute offspring to the next generation, while other females had a higher number of descendants.For minimum weighted coancestry (mFm), $Fx=\sum^{}m_{j}C_{jk}m_{k}$, where *m_j_* (*m_k_*) is the mean coancestries between the individual *j* (*k*) and all the other animals in the mating group. This strategy is employed to account for the current representation of each individual in the group and penalizes those already highly represented. Two alternatives were checked:Strategy mFm0, i.e., not all males were expected to contribute offspring to the next generation;Strategy mFm1, i.e., all the males were expected to contribute offspring to the next generation.For minimum weighted increase in coancestry (ΔmΔFΔm), $Fx=\sum^{}\Delta m_{j}\Delta C_{jk}\Delta m_{k}$, where Δ*m_j_* and Δ*m_k_* are the mean of the increase in coancestries [19] between the corresponding individuals and all the other animals involved in the mating plan. This strategy simultaneously accounts for the possible differences in pedigree among the animals, and their current representation, again with two options:Strategy ΔmΔFΔm0, i.e., not all males were expected to contribute offspring to the next generation;Strategy ΔmΔFΔm1, i.e., all the males were expected to contribute offspring to the next generation.


### 2.3. Strategies to Minimize the Offspring Coancestry


6.For minimum offspring coancestry (C), $Fx=\sum^{}C_{lm}$, where *C_lm_* is the coancestry between two offsprings *l* and *m* of the mating design. Each mating was assumed to provide exactly one offspring for the next generation and two alternatives were also checked:Strategy C0, i.e., not all males were expected to contribute offspring to the next generation;Strategy C1, i.e., all the males were expected to contribute offspring to the next generations.7.For minimum offspring increase in coancestry (ΔC), $Fx=\sum^{}\Delta C_{lm}$, where *ΔC_lm_* is the increase in the coancestry [15] between two offspring *l* and *m* of the mating design. Each mating only provides exactly one offspring, and also with two alternatives:Strategy ΔC0, i.e., not all males were expected to contribute offspring to the next generation;Strategy ΔC1, i.e., all the males were expected to contribute offspring to the next generations.8.The strategy for minimum offspring coancestry free (Cf) was similar to Strategy 6 (C), but, in this case, not all the females were expected to contribute offspring to the next generation, while other females had a higher number of descendants.9.The strategy for minimum offspring increases in coancestry free (ΔCf) was similar to Strategy 7 (ΔC) but, in this case, not all the females participated with offspring for the next generation.


### 2.4. Mixed Strategies


10.The strategy for mixing information from two generations (mix) was a combination of Strategies 1 and 6, i.e., $Fx=p_{1}\sum^{}C_{jk}+\left(1-p_{1}\right)\sum^{}C_{lm}$. All terms in the expression are already given above, being *p*_1_ a value between 0 and 1 indicating the weighting to be given in the coancestry in the previous generation. These values were tested, and all females were expected to contribute at least one offspring to the next generation, but not all males. The following values were tested for *p_1_*: 0.01 (mix 1–99), 0.05 (mix 5–95), 0.50 (mix 50–50), and 0.95 (mix 95–5).11.The strategy for mixing information from two generations free (mixf) was similar to Strategy 10 (mix) but, in this case, not all males and all females were expected to have an offspring for the next generation, while other females had a higher number of descendants. These scenarios were called mixf 1–99, mixf 5–95, mixf 50–50, and mixf 95–5.


### 2.5. Computations Performed

One hundred replicates of all strategies were conducted during 20 discrete generations. Each generation was formed by a random number of animals simulated from a Poisson distribution with the same size of the reference population to be mated. The generations had on average the same rate between males and females and randomly assigned the sex of the born animals.

For the sake of simplicity, the performance of the different strategies was evaluated only by the evolution of the effective population size [18].

Modifications were done on Endog v4.9 [27] to carry out all the simulations. Endog was also used to calculate the effective population size obtained by the different strategies, and this version is available for users upon request.

## 3. Results

The data structure of both reference populations used as the starting point in the simulations are summarized in Table 1. The pedigree completeness was similar in both reference populations and the sexual ratio was higher in the Almería population. The La Lajita population was more inbred than the Almería population and presented lower effective size and lower generation interval; both populations were not structured with a ratio NeC/NeF close to one.

The results of the simulations are compiled in Appendix A, that is, all the numeric results of the effective population size evolution throughout the generations obtained from each strategy in both populations. In this section, only those results that are necessary to illustrate the performance of the strategies tested are given.

### 3.1. Almería Population

Figure 1 shows the Ne in the simulated generations, i.e., 1 (short term), 5 (medium term), and 20 (long term) for all methods in the Almería population. In the short term, the best strategy was Ff, followed by mixf 95–5 and mixf 50–50. In contrast, the worst performances in the first generation were obtained using Strategies C0, C1, ΔC0, and ΔC1.

The results in the fifth generation show that strategies based on parents coancestry (F) lost the advantage observed in the first generation in favor of mixed methods giving equal or more weight to the minimization of offspring coancestry, thus, maintaining the advantage. The strategies based on minimization of offspring coancestry did not lead to more diversity in the generation.

The mixed strategies with more or equal weight on minimizing the offspring coancestry gave higher effective population size in the long term, in particular for the mixf 5–95 and mix 5–95 strategies. The strategies that gave higher losses of genetic variability in the long term were Ff but mainly mFm0.

Figure 2 shows the evolution of Ne by generation across methodologies. The performances of the mixf 50–50 or mixf 5–95 strategies are outstanding both in the short and long term. In contrast, the Ff and mix 95–5 strategies had a significant drop in effective population size from the fifth generation.

Nevertheless, among the methods that minimize the coancestry of the offspring, the strategies that all males and all females were not restricted to participate (Cf and ΔCf), and the strategies where only the females were restricted to participate with at least one descendant (C0 and ΔC0), had slightly better performance in the long term than strategies that restricted all females and males to participate (C1 and ΔC1).

In all the mixed strategies, the males had no restrictions on the expected number of descendants (mix and mixf), but in some strategies, all females had to participate (mix). In this group, strategies in which females did not have to contribute to the next generations were better in the short term. However, in the long term, the participation of all females was better in all the mixed strategies, except for mixf 5–95 and mix 5–95, where mixf 5–95 was better.

Finally, by comparing the group of strategies that minimize coancestry between the parents, the evolution of effective population size throughout the generations was similar for the ΔF0 and ΔF1 strategies, as well as the F0 and F1 strategies, in both the short term and the long term. This can be seen from observing the effect of minimizing increases instead of the coefficients themselves, as seen in Figure 1 and Figure 2. Within the methods minimizing the coancestry of the offspring, the C0, C1, and Cf strategies achieved greater effective population size than the ΔC0 and ΔC1 strategies in the long term. In the long-term, the mFm1 strategy was better than the ΔmΔFΔm1 strategy. However, the ΔmΔFΔm0 and mFm0 strategies performed similarly in both the short and the long term.

### 3.2. La Lajita Population

Figure 3 shows the variation in effective population size in Generations 1, 5, and 20 obtained by the different strategies in the La Lajita population. Figure 4 shows the evolution of the effective population size obtained by the different strategies throughout 20 generations in the La Lajita population. The methods with optimized effective population size were the mixf 95–5 and Ff strategies in the short term. The ΔC1 and C1 strategies had the worst performance at this stage. The best performance for this reference population in the final generations was assessed for the mixf 5–95 strategy, followed by the mix 5–95 strategy. As shown in Figure 3 and Figure 4, the strategies that gave higher losses of effective population size in the final generations were Ff and ΔmΔFΔm0.

The methods that minimize the coancestry between the parents performed worse in the early generations when all the males and females had to participate (F1, ΔF1, mFm1, and ΔmFΔm1) than the strategies when only females had to participate (F0, ΔF0, mFm0, and ΔmFΔm0) (see Figure 3 and Figure 4). The F1, ΔF1, mFm1, and ΔmFΔm1 strategies were also worse than the Ff strategy, when neither all males nor all females had to contribute to the next generations. However, the effective population size significantly dropped after the first generation using this strategy. In any case, the strategies when all females and all males were expected to have offspring for the next generations were better in the long term than the others in this group of strategies.

Among the strategies employed to minimize the offspring coancestry, when all females but not all males were restricted to participate (C0 and ΔC0), or when none of them were restricted to participate (ΔCf and Cf), the effective population size was maximized in the final generations, unlike C1 and ΔC1, with both sexes restricted to participate, which were worse in the long term.

All the methods with more weight on minimizing the coancestry of the offspring were better in the short term when all the females were not expected to participate. Moreover, methods giving some weight to the minimization of the parents’ coancestry were better in the short term when males and females were not forced to contribute to the following generation. The fact that all females contributed with at least one descendant in the next generation improved the effective population size in all mixed strategies in the long term.

Regarding the effect of minimizing increases instead of the coefficients themselves, first of all, the ΔF0, F0 and ΔF1, F1 strategies were similar in the long term. Moreover, the mFm0, ΔmΔFΔm0 and mFm1, ΔmFmΔ1 strategies were also similar, but the ΔC0, ΔC1, ΔCf and C0, C1, Cf strategies stayed similar over all generations.

## 4. Discussion

It has been widely reported that the best method to maintain or maximize the genetic diversity in a population is by minimizing coancestry among the offspring [8,9,31]. Whenever possible, this method should converge with the traditional methods of minimizing variance family size, maintaining similar numbers of females and males, and constant maintenance of the population census [9,10]. However, most studies hardly fit to real breeding scenarios. In this study, the performances of various strategies were assessed under more realistic conditions, i.e., all reproductive individuals of each sex or both sexes may not contribute offspring to the next generation. Moreover, the simulations were designed to ascertain the effect in the short, medium, and long terms of the strategies aimed at minimizing the coancestry of the offspring versus other strategies aimed at minimizing the coancestry of the parents or, in a mixed approach, giving different weights to coancestry of the parents and offspring. Moreover, for the first time to the best of our knowledge, the usefulness of new methods based on individual increase in inbreeding or on pairwise coancestries were analyzed to compute the global increase in inbreeding between cohorts [18,19]. This study was carried out using the genetic background of two subpopulations as the starting point. This enabled us to check the influence on performance of different initial conditions such as particular genetic structure, sex ratio (females/males), effective population size, and other factors naturally present.

When the aim was to minimize the coancestry between the parents to be mated, both Ff and mixf 95–5 were the best strategies with the best performance in the short term. However, these strategy results in a population of full- and half-sib families in the first generation of the breeding plan because these methods are focused in minimizing the inbreeding in the offspring and the methods select only a few couples to have offspring. Consequently, a great loss of genetic variability was expected from the second generation onwards [9,32]. The strategy mixf 95–5 was basically the same as Ff but selection of the coancestry among the offspring was weighted at a low 5%. This small use of offspring coancestry dramatically affected performance, i.e., the mixf 95–5 strategy performed well in the first generation and was also favourable in the middle and the long term as comparing with the Ff strategy. The other strategies minimizing the coancestry between the parents (F0, F1, ΔF0, ΔF1) and between the parents and all other animals involved in the mating (mFm0, mFm1, ΔmΔFΔm0 and ΔmΔFΔm1) were less effective in both the short and the long term. In this group of strategies, the methods using all females, all males, or all reproductive individuals of both sexes as parents of the next generation (F1, mFm1, ΔF1 andΔmΔFΔm1) always performed better in the long term than the strategies with no restrictions on the number of males involved in matings, (F0, mFm0, ΔF0 and ΔmΔFΔm0), or with any restrictions at all (Ff). The use of all reproductive females kept some additional genetic variability leading to a better performance in the long term.

When strategies aimed at minimizing the coancestry of the offspring were considered (C0, C1, ΔC0, ΔC1, Cf and ΔCf), the present study departed from previous studies in the literature [8,9,31]. In this group of strategies, the long-term effect on Ne of using all females, all males, or all reproductive individuals of both sexes as parents for the next generation was the opposite, when the aim was to minimize the coancestry between the parents. The Ne in this group of methods was not optimized by placing no restrictions on the number of mating animals.

Regarding mixed methods, the strategies giving the same or higher weight to the coancestry of the offspring (mixf 1–99, mixf 5–95, mixf 50–50, mix 1–99, mix 5–95, mix 50–50) were always among the best in the long term in both the Almeria and the La Lajita populations. Such strategies performed better than those giving more weight at minimizing the parents coancestry (mixf 95–5 and mix 95–5). Mixed strategies were superior to those focusing on offspring coancestry only (C0, C1, ΔC0, ΔC1, Cf, and ΔCf), i.e., weighting 1%, 5%, or 50% in minimizing coancestry between parents improved for both the Almería and La Lajita populations in the short term, as expected, but also in the long term. Strategies aimed at minimizing coancestry were not expected to structure the population in families. Nevertheless, it could be expected that weighting 1%, 5%, or 50% in minimizing coancestry between parents could lead to structuring in some families [9,15]. However, NeC/NeF kept stable throughout generations [19]. There is empirical evidence in real populations suggesting that minimizing coancestry between mating individuals balances allelic frequencies at the neutral marker level across years and generations, thus, avoiding losses of gene diversity [32]. Finally, restrictions on the number of reproductive individuals that contribute to the next generation had a slight but differential effect in each of the two populations, and highlighted the fact that different starting scenarios may influence the performance of the methodologies to be applied.

We also tested the usefulness of methods using increases in pairwise coancestry rather than absolute values of the coancestry to implement strategies to optimize Ne in small captive populations. However, we could not assess a different pattern between these two methods in the two populations, this might be due to a balanced pedigree depth of the individuals included in the reference population at the starting point. More tests are needed to establish the characteristics of the population structure that might limit the theoretical advantages of these methods before they are implemented.

Furthermore, the strategy of choice in both populations should be mixf 5–95. The greatest effective population size in the early generations was obtained by the mixf 50–50 strategy, which could ensure the viability of the population during a critical period for the following generations. However, the mixf 5–95 strategy produced a significant improvement in the effective population size in short term and also produced the largest effective size in both the middle and long terms. Minor differences have been found between the studied subpopulations because they are too much similar in management. However, it was observed that the starting population determined the positive or negative tendence of the stategies throughout generations. Therefore, these differences suggest that the strategy of choice should have to be studied for each case.

Hence, if all the variables were analyzed in both populations, weighting less or equal on coancestry of the parents had benefits in the short term. These methods were also amongst the best at maintaining an effective population size in the long term.

In any case, if the goal is to ensure a higher Ne in the long term, mixed methodologies must be applied by using coancestry of both parents and offspring with different weightings. If there is need for a high Ne, even in the early generations, a test must be carefully applied before implementing it in a preservation program.

Although this study focuses on concerns affecting the performances of different strategies aimed at optimizing Ne in small captive populations at the population level, it is worthwhile mentioning that there are concerns also at the species level. Not all the species have the same generation interval, and this must be taken into account. This parameter is mainly determined by the ecology and behaviour of the species but may be shortened due to the implementation of selection programs. It may even be unwittingly lengthened in conservation programs because the loss of genetic variability occurs per generation. In these gazelle populations, their age varied from 4.88 (La Lajita) to 6.36 years (Almería). However, in other species, such as the horse, the generation interval could be up to 12 years [33], while species such as rabbit may have a generation interval of about one year [34]. In populations with a longer generation interval, the main objective of a conservation program would be to keep the genetic variability as much as possible to ensure the survival of the species in a reasonable period of time. Furthermore, it might not be expected that the population remains closed for so long. Therefore, the implementation of a breeding program aimed at maximizing the effective population size in the short or medium term might be preferable than the application of a program in which the maximum potential is reached from Generation 20. On the contrary, in populations with a smaller generation interval, minimizing Ne in the long term might be more appropriate and affordable. Other conditions should be taken into account in practical scenarios and for future implementation of software about mating designs in conservation programs. For example, Gazelle species are polygynous [35]. As such, in nature, not all males reproduce. Females choose who to mate with from all the males available to reproduce during the mating season. Female choice is a behavioural mechanism by which females maximize their reproductive success and can potentially obtain direct or indirect genetic benefits for their progeny as it better reflects the natural (real) situation of these species in wild. Female mate choice is a non-random mating and this has correlated demographic effects, one of those being the potential to reduce the effective population size [36] by causing the sex-specific distributions of individuals (males) that successfully reproduce to diverge. Thus, it should be noted that the simulations have been carried out using only these pedigrees as a real starting point, but without considering all specific natural features of this specie. For example, in gazelles, from an evolutionary point of view, perhaps there is no reason to use all males for mating in these species as they are probably very well adapted to this mechanism of sexual selection.

## 5. Conclusions

From the present study, it is clear that, when concerns regarding real world populations are taken into account, there is no universal selection strategy that can be used to optimize Ne in any small captive population in the short, medium, and long terms. Although these simulations identified patterns useful for many scenarios, clearly there were some differences in the performance of the strategies applied in the two populations used for testing. This is important because our two reference populations belong to the same species (*Gazella cuvieri*) and have very similar structure and pedigree information. Other more variable scenarios may give higher differences. Moreover, each population had different real conditions, for instance, the availability of females and males for breeding. Therefore, each particular population may have its optimal mating strategies and that might have to be carefully studied before incorporating it in a breeding plan.

In any case, it can be concluded that the strategies aimed at minimizing the coancestry between the parents were beneficial in the short term but could be detrimental in the medium and long terms. In turn, strategies aimed at minimizing the coancestry among the offspring cannot be considered to be selection strategies in the short term or in the long term. Strategies that give equal or more weight to the minimization of the coancestry of the offspring than to the minimization of the coancestry of the parents were preferable to other strategies in both the short and the long terms. Depending on the circumstances, sometimes it is advisable to restrict mating plans requiring all available females and/or males to have an offspring for the next generation that in the long term might have beneficial effects.

## Figures and Tables

**Figure 1 animals-11-01559-f001:**
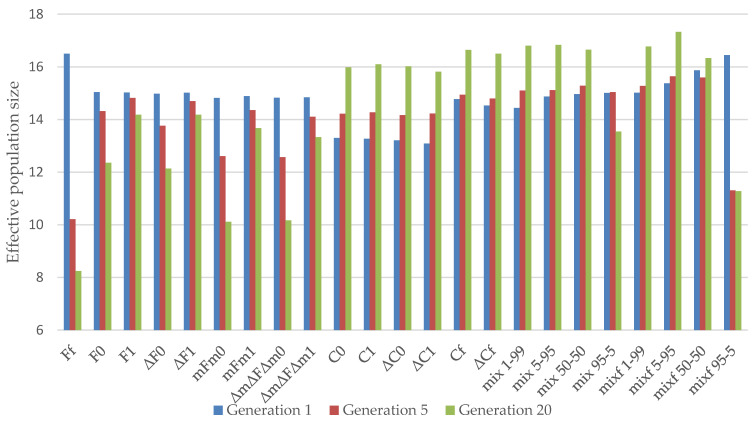
Effective population size obtained by the different strategies in the reference population of Almería in 1, 5, and 20 generations. (Ff, F0, and F1) based on minimum coancestry of the parents; (ΔF0 and ΔF1) based on increment in coancestry of the parents; (mFm0 and mFm1) based on weighted coancestry; (ΔmΔFΔm0 and ΔmΔFΔm1) based on weighted increase in coancestry; (C0, C1, and Cf) based on coancestry of the offspring; (ΔC0, ΔC1, and ΔCf) based on increase in coancestry of the offspring; (mix 1–99, mix 5–95, mix 50–50, mix 95–5, mixf 1–99, mixf 5–95, mixf 50–50, and mixf 95–5) mixing information from the parents’ coancestry and offspring’s coancestry.

**Figure 2 animals-11-01559-f002:**
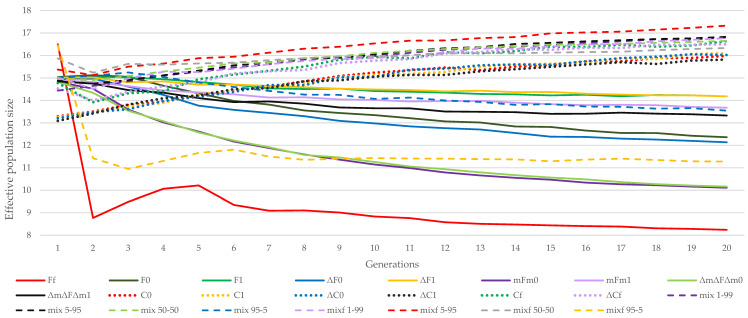
Evolution of effective population size obtained by the different strategies in the reference population of Almería throughout 20 generations. (Ff, F0, and F1) based on minimum coancestry of the parents; (ΔF0 and ΔF1) based on increment in coancestry of the parents; (mFm0 and mFm1) based on weighted coancestry; (ΔmΔFΔm0 and ΔmΔFΔm1) based on weighted increase in coancestry; (C0, C1, and Cf) based on coancestry of the offspring; (ΔC0, ΔC1, and ΔCf) based on increase in coancestry of the offspring; (mix 1–99, mix 5–95, mix 50–50, mix 95–5, mixf 1–99, mixf 5–95, mixf 50–50, and mixf 95–5) mixing information from the parents’ coancestry and offspring’s coancestry.

**Figure 3 animals-11-01559-f003:**
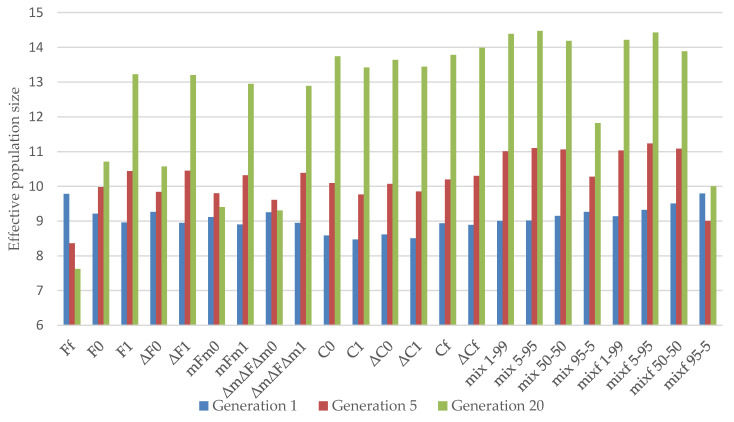
Effective population size obtained by the different strategies in the reference population of La Lajita in 1, 5, and 20 generations. (Ff, F0, and F1) based on minimum coancestry of the parents; (ΔF0 and ΔF1) based on increment in coancestry of the parents; (mFm0 and mFm1) based on weighted coancestry; (ΔmΔFΔm0 and ΔmΔFΔm1) based on weighted increase in coancestry; (C0, C1, and Cf) based on coancestry of the offspring; (ΔC0, ΔC1, and ΔCf) based on increase in coancestry of the offspring; (mix 1–99, mix 5–95, mix 50–50, mix 95–5, mixf 1–99, mixf 5–95, mixf 50–50, and mixf 95–5) mixing information from the parents’ coancestry and offspring’s coancestry.

**Figure 4 animals-11-01559-f004:**
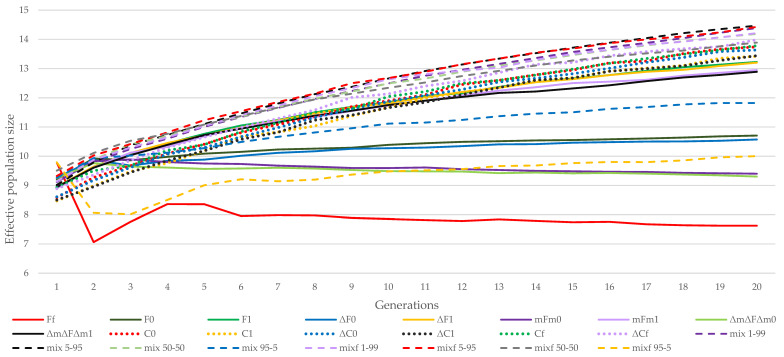
Evolution of effective population size obtained by the different strategies in the reference population of La Lajita throughout 20 generations. (Ff, F0, and F10 based on minimum coancestry of the parents; (ΔF0 and ΔF1) based on increment in coancestry of the parents; (mFm0 and mFm1) based on weighted coancestry; (ΔmΔFΔm0 and ΔmΔFΔm1) based on weighted increase in coancestry; (C0, C1, and Cf) based on coancestry of the offspring; (ΔC0, ΔC1, and ΔCf) based on increase in coancestry of the offspring; (mix 1–99, mix 5–95, mix 50–50, mix 95–5, mixf 1–99, mixf 5–95, mixf 50–50, and mixf 95–5) mixing information from the parents’ coancestry and offspring’s coancestry.

**Table 1 animals-11-01559-t001:** Structure data of the reference populations of Almería and La Lajita (EG, equivalent complete generation; NeF, effective population size based on individual increase of inbreeding; NeC, effective population size based on increase of coancestry).

	Population of Almería	Population of La Lajita
Males	6	8
Females	16	11
Ratio females/males	1.80	1.30
EG	8.78	9.01
Standard deviation of EG	0.64	1.04
Inbreeding	0.26	0.37
Inbreeding of males	0.27	0.37
Inbreeding of females	0.24	0.37
NeF	13.27	8.91
NeC	12.90	8.14
NeC/NeF	0.97	0.91
Generation interval	6.36	4.88

## Data Availability

The studbook of the *Gazella cuvieri* is available at http://www.eeza.csic.es/documentos/Studbook_G%20cuvieri%202019.txt (accessed on 25 May 2021).

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
