# Peer review of "Breeding Strategies to Optimize Effective Population Size in Low Census Captive Populations: The Case of Gazella cuvieri"

_animals, 2021, doi:10.3390/ani11061559_

Round 1
Reviewer 1 Report
The objective of the article is to measure the effect on effective size at midterm and long term of various conservation strategies based on real pedigree data from two strains of a wild endangered species.
The objective of the article are quite interesting and the results could be very relevant for captive populations or rare breeds’ managers. However, substantial work is needed to shorten the article and pinpoint to the most relevant results and discussion, in a clearer way. The discussion part is quite weak and not referring to other type of work about the topic, for instance in wild species and species with a short generation interval (mouse, rabbit, chicken…). The general feeling is that the article is a transcript of an oral presentation. The text also needs to be reviewed extensively for English language.

Author Response
Dear Reviewers,
We are sending a revised version of the manuscript animals-1197306. We are grateful for your comments. The comments have enabled us to improve the quality of the paper. All the changes are in red along the text. The lines in the cover letter refer to the final version in word.
Best regards
Reviewer 1.
The objective of the article is to measure the effect on effective size at midterm and long term of various conservation strategies based on real pedigree data from two strains of a wild endangered species.
The objective of the article are quite interesting and the results could be very relevant for captive populations or rare breeds’ managers. However, substantial work is needed to shorten the article and pinpoint to the most relevant results and discussion, in a clearer way. The discussion part is quite weak and not referring to other type of work about the topic, for instance in wild species and species with a short generation interval (mouse, rabbit, chicken…). The general feeling is that the article is a transcript of an oral presentation. The text also needs to be reviewed extensively for English language.
The introduction is too long and needs to summarize in a more intelligible way what are the issues and objectives of the study.
ANSWER: We are grateful for the comment of Reviewer 1. The introduction and discussion section have now been shortened as suggested. Some references have been added related with this topic in farm animals as suggested below ([4],[11],[12]) The English has been revised by a native speaker, Lawrence Baron (lawrencebaron@gmail.com).
Some specific comments.
49/50 “When dealing with small size populations, researchers in animal breeding have paid major on mating strategies to maximise genetic gain under artificial selection pressure whilst minimizing inbreeding and loss of genetic variation”
This is true for breeds that have sufficient numbers to perform selection pressure, how a great majority of small size populations are just running conservation programs where the only objective is the management of genetic variability (see for instance Gicquel E, Boettcher P, Besbes B, Furre S, Fernández J, Danchin-Burge C, Berger B, Baumung R, Feijóo JRJ, Leroy G. Impact of conservation measures on demography and genetic variability of livestock breeds. Animal. 2020 Apr;14(4):670- 680. doi: 10.1017/S1751731119002672.). Indeed, programs maximizing genetic gain and genetic variability are used mostly in large breeds but with a small effective population size (see for instance Granleese T, Clark SA, Swan AA, van der Werf JH. Increased genetic gains in sheep, beef and dairy breeding programs from using female reproductive technologies combined with optimal contribution selection and genomic breeding values. Genet Sel Evol. 2015 Sep 14;47(1):70) ; So, in my view, what zoos are dealing with is exactly the same as what rare breeds managers have to tackle.
ANSWER: We thank the Reviewer 1 for pointing this out. The paragraph has been modified following his/her suggestions and new references have been added (L 49-52).
58/59 There is consensus on the fact that minimizing coancestry (f) in a cohort [6] outperforms other breeding strategies [7][8].
The consensus is shown by giving two references from the same authors. I agree that they are major actors in the field, but may be a reference from another team could be given?
ANSWER: We have added these two references: [11],[12].
59/60: rephrase, it is difficult to grasp all the concepts with a single sentence.
ANSWER: The sentence has been rephrased [L 59-63]
62/63: the verb is missing ?
ANSWER: We removed this sentence to shorten the Introduction section as much as possible.
64/67: long sentence for not such an easy concept, use of commas and/or breaking the sentence in two would be advisable.
ANSWER: We thank the Reviewer 1 for pointing this out. We have clarified the sentence (L64-67).
67/69 Furthermore, in such a breeding programme, close relatives tend to bee mated and, in the medium and long term, this leads to populations structuring and the formation of inbred subpopulations.
ANSWER: We removed this sentence to shorten the Introduction section as much as possible.
73/74 Erase or rephrase (“complain”)
ANSWER: We removed this sentence to shorten the Introduction section as much as possible.
76/78: Rephrase (“they cause”)
ANSWER: This has been clarified (L72-73)
80/82 Erase or rephrase (“real world”)
ANSWER: This has been clarified (L75-76).
-84/85: Rephrase (“once again”)
ANSWER: This has been clarified (L78-79)
86/87: new methods have been developed to do what ?
ANSWER: This has been clarified (L79-82).
90/94 rephrase and summarize. I didn’t understand point b
This has been clarified (L83-88).
90/91 a) the starting point of the simulations hardly fit an ideal scenario including unrelated individuals only; ð I think it is the opposite that is true, which is real life situations do not fit the simulations hypothesis most of the time, for instance individuals are often related?
ANSWER: We apologize for the misunderstanding, we want to say that “the ideal situation that all the individuals are unrelated” is difficult to accomplish in real life. This has been clarified (L83-88).
92/93: mating policies => breeding schemes ?
ANSWER: This has been modified as suggested (L89-90)
96/97 The present research compares computer simulations of the performance of different breeding schemes aimed at limiting the loss of genetic diversity in small size captive populations. ð I think it can be interesting for any small size populations;
ANSWER: This has been modified as suggested (L89-90).
105/106: define more precisely what is the reference population. For instance: the reference population of the la lajita flock consists of all the individuals born between 2014 and 2015 (11 females and 8 males). Reference 19 is not available and the concept is widely known, may be switch with an available article. Results of the material needs to be put in the results section (table 1).
ANSWER: The reference population has been defined more precisely (L111-113). The [19] reference has been modified by reference [28]. Table 1 has been moved to results.
121 Several mating were essayed. => Inappropriate
ANSWER: This has been modified (L124)
124/126: the strategies are presented in short then in much longer way right after which is redundant. The short presentation should be skipped.
ANSWER: This has been deleted.
126 “complex strategies” => complex = that cannot be explained. Change of word is suggested.
ANSWER: The word complex has been changed by mixed.
137/139: different pedigree depth of animals was considered important => define important and coancestries were increased per generation
ð Increased how?
ANSWER: The text has been clarified: (L138-139)
“Not all males were expected to have offspring for the next generation” and “ All the males had at least a female to mate with” => rephrase
ANSWER: This has been modified in methods’ section
237 Regarding the effect of all males, all females or both contributing with offspring to the following generations => rephrase
ANSWER: We removed this sentence to clarify the Results section as much as possible.
Results sections: I’m not sure it is advisable to divide the results between each flocks; the most striking results and differences should be stated in this part, with reference to the flock, in order to shorten this part.
ANSWER: We have carefully considered the suggestion of Reviewer 1. However, since one of the goals of the work is to highlight the fact that different starting points may lead to the implementation of different breeding programmes in different populations we have, respectfully, considered it necessary to keep the current structuring of the Results section in two different subsections.
314/315 that all reproductive individuals of each sex or both sexes may not contribute or not with offspring to the next generation.
ANSWER: This has been modified as suggested (L321-322).
319 Furthermore, for the first time in the literature we tested the usefulness => rephrase
ANSWER: This has been modified (L326-327)
322/323 sex rate =>sex ratio
ANSWER: This has been modified (L 329)
327 – paragraph : This paragraph presents mostly results; this should be summarized and focused on the outputs for discussion. Also, the relevance of dividing the discussion on a specific item (which is mostly presenting results) then the general discussion is questionable.
ANSWER: Thank you for your suggestion. The Discussion section has been shortened as much as possible to focus on the output for discussion and we have removed the division of the discussion section.
373: “For the first time in the literature” = delete
ANSWER: This has been deleted.

Reviewer 2 Report
Dear Authors,
General Comments:
I believe it is an interesting and important work to the conservation of this wild specie and other wild species at risk. Simulation is a good tool to assist decision-making, mainly in extreme context as this of Gazella sp. The simulated strategies were well designed, covering important aspects of animal conservation. I missed some details about this specie in nature, particularly about the population size and structure, family structure, and reproductive behavior. It would help you and us to better understand how realistic was the simulation design and to discuss about the results obtained. There is lack of discussion about genetic aspects involved in strategies to minimize the impact on the effective size of this very small population, e.g., on the consequences of each strategy to the genetic variability between and within families. It would also be interesting to consider or hypothesize n all the parameters presented in Table 1 in the discussion section, mainly the sex ratio, because of the results when females were restricted or not. The discussion has mainly compared the strategy impacts on Ne. I suggest you to improve it. Your effort deserve this improvement. How was the choice of individuals to be parents of the next generation? The manuscript needs some English language revision. I will give some examples below. Congratulations for the work.
Specific comments:
Line 45: Which conditions?
Line 46: Standardize the keywords section according to the Journal rules. Comma or semicolon?
Line 51: Is it exactly what the references cited said?
Lines 54-55: Writing suggestion: Management of populations for preserving purposes need...
Lines 64-68: Revise English orthography and grammar. ... in short term to bee... Are you speaking about bees?
Lines 68-69: What does it means? I couldn't understand.
Lines 79-80: Please, justify this comment? Are you sure of it? F-statistics is nowadays, in genomic era, an accessible parameter to study population structure and diversity.
Line 91: English language revision. ... "may giving"... by "may give"
Line 103: ...using as a starting point two subsets... I miss something here. Suggestion: ...using as a starting point information from two subsets...
Line 106: Remove the second "and". ...females and 6 males born between...
Line 286: English language revision: "expexted" by "expected".
Figure 4: Some of the curves in legend are missing.
Lines 319-320: Are you sure about this affirmative? For the sake of safety, I suggest you to insert "to our knowledge".
Lines 335-336: If all of them are favorable, "but" is not necessary and sounds like a controversy at first. Simplify.
Lines 338-341: ... was never the strategy of choice... Have you or someone else choose this strategy before the simulation? What does it means? Clarify.
Lines 375: Again. Insert " to our knowledge".
Line 377: What did you mean by "standard differential pattern"?
Author Response
Dear Authors,
General Comments:
I believe it is an interesting and important work to the conservation of this wild specie and other wild species at risk. Simulation is a good tool to assist decision-making, mainly in extreme context as this of Gazella sp. The simulated strategies were well designed, covering important aspects of animal conservation. I missed some details about this specie in nature, particularly about the population size and structure, family structure, and reproductive behavior. It would help you and dobus to better understand how realistic was the simulation design and to discuss about the results obtained. There is lack of discussion about genetic aspects involved in strategies to minimize the impact on the effective size of this very small population, e.g., on the consequences of each strategy to the genetic variability between and within families. It would also be interesting to consider or hypothesize n all the parameters presented in Table 1 in the discussion section, mainly the sex ratio, because of the results when females were restricted or not. The discussion has mainly compared the strategy impacts on Ne. I suggest you to improve it. Your effort deserve this improvement. How was the choice of individuals to be parents of the next generation?The manuscript needs some English language revision. I will give some examples below. Congratulations for the work.
ANSWER: We are grateful for the referee’s comment. We have included details about the specie in nature (L97-108) and some discussion has been added about reproductive behaviour limitations in wild populations (L408-419). Also the discussion has been enriched by clarifying the consequences of each strategy (L364-366). Also the method section has been clarified in order to explain how is the choice of an individual to be parents that is randomized inside the restriction of each method in particular (199-202).
The English has been revised by a native speaker, Lawrence Baron (lawrencebaron@gmail.com).
Regarding the parameters in table 1, they describe the populations at the starting point, the simulations do not change the initial ratio in the simulated scenarios. As the software is freely available. This is a general strategy of the program to make it useful to other researchers with their own dataset thinking that the best ratios to be used are those already present in the real data. This could be interesting, but it would increase the number of scenarios considerably and this was not our goal when we designed the study, but we think this is an important point and it has been discussed in the manuscript (L413-417).
Specific comments:
Line 45: Which conditions?
ANSWER: This has been clarified (L44-45).
Line 46: Standardize the keywords section according to the Journal rules. Comma or semicolon?
ANSWER: This has been clarified (L46).
Line 51: Is it exactly what the references cited said?
ANSWER: We agree with referee and we have changed this. In fact, optimal contributions strategies used to be implemented in large breeds with low effective population size. Nevertheless, in small size populations the aim is to preserve genetic diversity. The paragraph has been modified following Reviewer 1’s suggestions and new references have been added (L49-52).
Lines 54-55: Writing suggestion: Management of populations for preserving purposes need...
ANSWER: We thank Reviewer 2 for his/her suggestion. However, we have deleted this sentence.
Lines 64-68: Revise English orthography and grammar. ... in short term to bee... Are you speaking about bees?
ANSWER: We thank Reviewer 2 for his/her suggestion. However, we have deleted this sentence in order to summarize as Reviewer 1 suggested.
Lines 68-69: What does it means? I couldn't understand.
ANSWER: It means that when strategies are based on minimising parent´s coancestry, in fact it minimises offspring inbreeding. Therefore, most of the offspring of the first generations are going to be selected from a few couples and then, the populations are going to be structured in inbred subpopulations of families. This has been clarified (L64-67)
Lines 79-80: Please, justify this comment? Are you sure of it? F-statistics is nowadays, in genomic era, an accessible parameter to study population structure and diversity.
ANSWER: We are sorry for the misunderstanding, we have clarified the sentence (L73-77).
Line 91: English language revision. ... "may giving"... by "may give"
ANSWER: We thank Reviewer 2 for his/her revision. However, we have deleted this to summarize as Reviewer 1 has suggested .
Line 103: ...using as a starting point two subsets... I miss something here. Suggestion: ...using as a starting point information from two subsets...
ANSWER: This has been clarified (L109-111).
Line 106: Remove the second "and". ...females and 6 males born between...
ANSWER: Removed as suggested.
Line 286: English language revision: "expexted" by "expected".
ANSWER: Corrected.
Figure 4: Some of the curves in legend are missing.
ANSWER: We have revised all the figures and the legends have been corrected.
Lines 319-320: Are you sure about this affirmative? For the sake of safety, I suggest you to insert "to our knowledge".
ANSWER: We agree with referee and we have inserted this (L327-328).
Lines 335-336: If all of them are favorable, "but" is not necessary and sounds like a controversy at first. Simplify.
ANSWER: Thank you for the comment. We have simplified this (L338-340).
Lines 338-341: ... was never the strategy of choice... Have you or someone else choose this strategy before the simulation? What does it means? Clarify.
ANSWER: We have deleted this. However, we want to infer that it is not advisable to select these strategies for the reference population.
Lines 375: Again. Insert " to our knowledge".
ANSWER: Thank you for the comment. However, we have deleted this to summarize as Reviewer 1 has suggested .
Line 377: What did you mean by "standard differential pattern"?
ANSWER: We apologized for the misunderstanding. We have clarified the sentence (372-378).
